# MASKED VAE: DISTRIBUTIONALLY-INFORMED SELF-SUPERVISED VISION LEARNING

## ABSTRACT

Masked pre-training with transformers is a popular self-supervised representation learning paradigm, initially showing success in natural language processing (NLP) before moving to computer vision (CV). However, an aspect of masked pre-training that is missing in CV is the ability to capture the distribution of possible outputs. In NLP pre-training methods, the distribution is expressed as a softmax output layer. In CV, the masked autoencoder (MAE) outputs a point estimate of the masked pixels' RGB values. This formulation is fundamentally limited, as it models an under-constrained problem as well-posed, leading to difficulties when deployed in cluttered scenes, which often contain occluded objects. It only provides one hypothesis for amodal completion, when in reality, occluded regions can often be completed in many different ways, *e.g.*, behind an open refrigerator door a scene could contain spoons, cups, pizzas, etc.. This inability to complete multiple modes indicates the weakness of the underlying representation in capturing contextual relationships. Towards creating a distributionally-aware formulation with contextually-aware representations, we propose the Masked VAE, a transformer-based self-supervised learning method that combines ideas from the MAE and the variational autoencoder (VAE). Additionally, we propose Context-Completion, a benchmark task that uses masked completion to measure the ability of representations to predict multiple potential objects in cluttered scenes. Evaluations show that our method outperforms other methods on Context-Completion, while matching state-of-the-art in representation performance in downstream classification tasks.

## 1 INTRODUCTION

Masked pre-training with transformers is a popular self-supervised representation learning paradigm. The idea first showed success in NLP, where masked pre-training methods like BERT (Devlin et al., 2018; Liu et al., 2019; Sanh et al., 2019) produced representations with state-of-the-art results in downstream tasks. This success inspired masked autoencoding (MAE) methods in computer vision (He et al., 2022; Feichtenhofer et al., 2022; Li et al., 2023; Wei et al., 2022), whose learned representations also achieved state-of-the-art results in downstream tasks.

However, the adaptation of masked pre-training to computer vision missed a crucial aspect of the NLP formulation: the ability to capture the distribution of possible outputs. In NLP methods like BERT, the output was a softmax probability distribution over potential token completions (Devlin et al., 2018). However, in vision MAEs, the output was a deterministic point estimate of the masked pixels' RGB values, instead of a distribution (He et al., 2022). That is, given visible tokens, the MAE always gives the same completion, even though there are often many possibilities. This is a fundamentally limited formulation, as it models this under-constrained problem as well-posed.

Standard evaluation methods do not consider the representations' ability to capture multiple hypotheses for a given image. While classification accuracy (in which the encoder is used as the classifier backbone) is one way to evaluate representation quality, it does not fully measure the learned contextual relationships. That is, given some objects in an image, we would expect a good representation to contain some notion of which objects are likely to co-occur. For example, if a partially occluded scene contains a cutting board, its representation should encode the possibility of utensils, fruits, etc. Luckily, the masked autoencoding framework gives us a way to evaluate this. After we

pass a partial image through the encoder, we can decode the representation into a full image. If the decoded full image contains contextually plausible inpaintings of the missing parts (*e.g.*, completes the ocean, given a beach), this is evidence that the encoder has learned a contextually-aware representation.

Towards a distributionally-aware formulation, we propose the Masked VAE, a transformer-based self-supervised learning method that combines ideas from the MAE and the variational autoencoder (VAE) (Kingma, 2013). Our key insight is to model the latent space tokens corresponding to masked pixels as samples from a multivariate Gaussian distribution, taking after the VAE. But, differently from the VAE, we only model masked tokens with this distribution – visible tokens are modeled deterministically, as in the MAE. Essentially, we replace the originally fixed masked tokens from the MAE with probabilistic latent codes like a VAE. Thus, when our Masked VAE inpaints masked samples given a partial image, it can use the deterministic latent space tokens from the visible portion, and sample latent space tokens for the masked portion, thereby allowing it to generate multiple possible completions. To ensure meaningful associations between such samples from the Gaussian distribution and real-space pixel values, during pre-training, we allow another encoder to "peek" at "masked" image patches and convert them to latent space tokens.

Furthermore, towards evaluating the contextual awareness of representations, we propose Context-Completion, a new benchmark task. Given masked images with partially visible context, a trained encoder-decoder method must complete the missing portions. The idea is that a better representation will lead to a distribution of inpaintings that adhere to the context of the scene. This task can be considered a visual analog of the popular cloze tasks in NLP (e.g., MadLibs). When evaluating the representations in this way, the masked portion should be a contiguous object, rather than randomly selected pixels, as a nearest-neighbor inpainting strategy could be reasonably productive in the latter case, whereas actual contextual knowledge must be leveraged to fill in a block. As such, this task uses the bounding boxes from the COCO (Lin et al., 2014) dataset. The deterministic nature of the standard MAE most obviously poses problems in these very contextual inference tasks. Ideally, it should produce conditional distributions (which should have entropy) of the hidden image tokens. If we have a partial kitchen scene, the masked-out portion could contain many plausible objects (*e.g.*, plates, cups, spoons). However, the current deterministic formulation will just inpaint the average pixel values, which are often semantically unmeaningful. A distributionally-aware formulation of the MAE could overcome this limitation by giving multiple semantically plausible inpaintings.

We investigate the implications of the design differences between our distributionally-aware Masked VAE and the standard MAE. In particular, we assess each method's ability in: (1) masked completion, *i.e.*, contextual inference; (2) classification via the self-supervised representations. Our Masked VAE is better at contextual inference, as measured by its ability to capture the conditional distribution of masked objects given context. Regarding the usefulness of the learned representations for classification, our Masked VAE is competitive with the standard MAE, and outperforms other leading methods, as evaluated via end-to-end finetuning. In summary, our contributions are as follows:

- We introduce the Masked VAE, a self-supervised learning method that builds upon the MAE by introducing distributional-awareness, in line with the ill-posed nature of the masked completion task.

- We propose Context-Completion, a benchmark test of representation quality that uses object inpainting as a way to measure the ability to capture contextual relationships.

- We investigate the properties of our Masked VAE and find that while distribution-awareness does not seem to be necessary for the representations to be useful for classification, it qualitatively and quantitatively helps a great deal in creating contextually plausible completions.

- Via end-to-end finetuning, we demonstrate that the representations learned by our Masked VAE are on par with leading methods for classification, without extra inference time cost.

## 2 RELATED WORK

### 2.1 VARIATIONAL AUTOENCODERS

The variational autoencoder (VAE) is a probabilistic graphical model that can be used for representation learning and generative modeling (Kingma, 2013). One of its distinguishing characteristics

is the adherence of the latent space to a multivariate Gaussian distribution, enforced by a Kullback-Leibler divergence (KLD) loss (Kullback & Leibler, 1951). Later, Higgins et al. (2017) introduced the beta-VAE, an adaptation of the VAE aimed at learning disentangled representations; this was achieved through the $\beta$ factor on the KLD loss. We build on these works by adopting the loss function from beta-VAE, but in our formulation not all latent factors are modeled by a multivariate Gaussian distribution.

## 2.2 MASKED PRE-TRAINING

Masked pre-training is a self-supervised learning paradigm in which a sequence, such as a sentence or an image, is tokenized, with certain tokens being masked out (*i.e.*, invisible). Then, a transformer model, given the visible tokens, must predict the values of the masked tokens. After this pre-training converges, the representations from the transformer encoder can then be used in downstream tasks (Vaswani, 2017; Devlin et al., 2018; He et al., 2022). This facilitates learning useful representations from vast quantities of unlabeled data, such that we can achieve better performance on downstream tasks with limited compute budgets or amounts of labeled data.

The idea originally gained prominence in the NLP community, with masked pre-training approaches like BERT achieving state-of-the-art performance in benchmark tasks (Devlin et al., 2018; Liu et al., 2019; Sanh et al., 2019). In the NLP formulation, the objective was to predict a softmax probability distribution over the *discrete* tokens that had been masked.

Later on, the idea was adopted in computer vision. BEiT (Bao et al., 2021) was one of the first such proposals; rather than directly optimizing for pixel reconstruction, they used DALL-E's discrete VAE (Ramesh et al., 2021) to encode images into discrete token space. After that, they conducted masked pre-training in the resultant discrete token space. In this discrete token space, they had a softmax objective, thereby achieving a form of distribution-awareness. However, one issue is that training the discrete VAE is non-differentiable, due to the discrete token space. Also, adopting a discrete token space may also not be the ideal formulation for vision problems, since vision data is inherently continuous (subject to machine precision). BEiT v2 (Peng et al., 2022) is a modification to the BEiT that uses a VQ-VAE (Van Den Oord et al., 2017) to generate the discrete token space; it suffers from similar limitations of non-differentiability in training the encoder to discrete space.

Later on, fully pixel-based masked autoencoders (MAE) (He et al., 2022; Wei et al., 2022; Feichtenhofer et al., 2022) with vision transformer backbones (Dosovitskiy, 2020) showed great success in representation learning. The objective for such MAE models was to predict the masked RGB pixel values, and did not make use of auxiliary models to convert any tokens to discrete space. At least as it relates to pixel-based methods, one aspect that seems to have been lost in translation from NLP to CV is the distributional awareness. In NLP masked pre-training, the distributional awareness was achieved by providing a softmax probability distribution over the possible tokens. For pixel values, there is not an immediately clear notion of a discrete softmax probability distribution, since pixel values are inherently continuous. This poses a problem, because one cannot know with certainty what the masked value should be. Thus, we hope to build a masked pre-training paradigm for vision that addresses this.

## 2.3 SELF-SUPERVISED VISION LEARNING

Besides masked autoencoding, other transformer-based approaches to self-supervised vision learning exist. One of the first was iGPT, which used an autoregressive pixel prediction task for pre-training (Chen et al., 2020a). iGPT's success was mostly confined to low-resolution images, and did not make use of ViTs (Dosovitskiy, 2020), which were a concurrent work. However, they were able to model the distribution of predicted pixels via discretizing the output space, either through k-means clustering or VQ-VAE (Van Den Oord et al., 2017). Even so, this does not capture the continuous nature of the vision modality. After ViTs were introduced, DINO (Caron et al., 2021) emerged, using a novel label-free knowledge distillation approach. MoCo-v3 (Chen et al., 2021) used a contrastive approach.

Another stream of SSL research builds upon the idea of maximizing representation similarity (either by direct comparison or via a prediction task) between two augmented views of the same image:

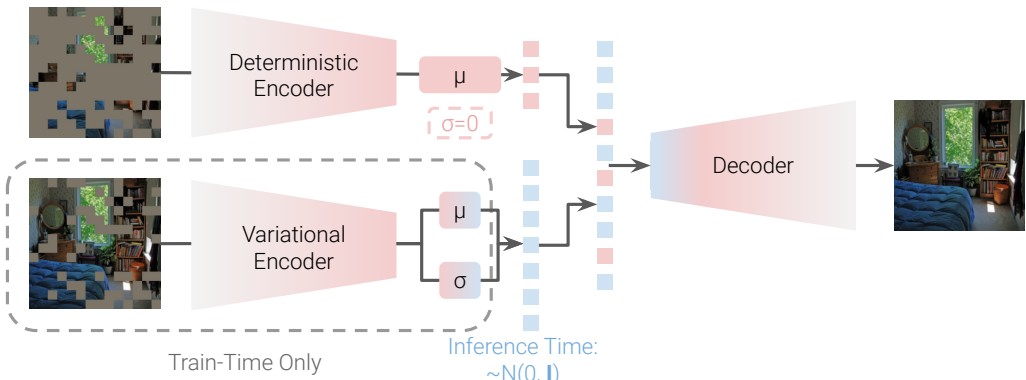

Figure 1: **Masked VAE System Diagram:** The input is a tokenized image. The deterministic transformer encoder processes one set of tokens to generate token-wise embeddings. Simultaneously, during train time, the variational transformer encoder processes the complementary set of tokens (*i.e.*, those that are not visible to the deterministic encoder), generating token-wise embeddings based on predicted means $\mu$ and variances $\sigma$ from a Gaussian distribution. These tokens are then combined via cross-attention into a transformer decoder to reconstruct the input image as output. During inference time, we discard the variational encoder and represent masked tokens with a standard Gaussian distribution.

this includes SimSiam (Chen & He, 2021), SimCLR (Chen et al., 2020b), BYOL (Grill et al., 2020), and Barlow Twins (Zbontar et al., 2021), SWaV (Caron et al., 2020).

Furthermore, there is a related stream of research that uses contextual inference as the self-supervision task (Doersch et al., 2014). Doersch et al. (2015) train networks to predict the relative positions of two patches from the same image. Noroozi & Favaro (2016) extend the idea by training networks to unscramble a jigsaw puzzle of the image. More recently, I-JEPA (Assran et al., 2023) used an approach that predicted the representation of an image block, conditioned on the representation of another block. Context Autoencoders (Chen et al., 2024) also predict the representations of image blocks, conditioned on the representations of other blocks, and take the extra step of reconstructing the masked blocks. Overall, this line of work in contextually-informed SSL informs our contextual inference-based evaluation.

## 2.4 CONTEXT-AWARENESS

We define context as plausible co-occurrence between groups of objects, features, and/or backgrounds. The importance of context in visual representation learning has been long-studied (Torralba, 2003). It is known that neural networks leverage contextual information to make predictions (Xiao et al., 2020; Kim et al., 2019; Lovering et al., 2021; Shetty et al., 2019; Winkler et al., 2019; Rosenfeld et al., 2018), and there are many lines of work that attempt to improve the context-awareness of models (Hu et al., 2018; Choi et al., 2011; Malisiewicz & Efros, 2009).

Madras & Zemel (2021) presented a formal framework for identifying out-of-context data. Their approach analyzed the co-occurrence statistics of different objects in the same image. To this end, they used the COCO (Lin et al., 2014) dataset, as it already had manually annotated object bounding boxes. We follow this line of work in designing our contextual inference benchmark (Section 4) by analyzing co-occurrence statistics from COCO bounding boxes.

## 3 ARCHITECTURE

Our architecture (see Figure 1) contains two primary branches: the visible tokens are processed as in the MAE (He et al., 2022), while the masked tokens' representations capture the distribution of values those tokens could take on.

### 3.1 Deterministic Encoder on "Visible" Tokens

The top branch in Figure 1 is the same as in the standard MAE. The deterministic ViT encoder (Dosovitskiy, 2020) processes the "visible" tokens, outputting token-wise embeddings. These tokens are then combined with the mask tokens and decoded to reconstruct the original image.

### 3.2 Variational Encoder on "Mask" Tokens

Where our method deviates from the original MAE is that rather than having a fixed mask token, we model the mask tokens as samples from a multivariate Gaussian distribution. In principle, taking different samples from the multivariate Gaussian distribution in the latent token space should allow the decoder to construct multiple possible completions of the partial image.

However, it is insufficient to naively replace the traditional mask tokens with Gaussian noise during the pre-training process, because then there would be no association between points in the multivariate Gaussian distribution and potential completions of the image. Taking inspiration from the VAE (Kingma, 2013), we add an additional encoder branch that is allowed to "peek" at the "masked" tokens in input pixel space during training time, such that it can meaningfully associate them with points in the latent space. We call this the variational encoder (bottom of Figure 1), and it has the exact same ViT architecture as the deterministic encoder. We add two parallel transformer blocks on top of this variational encoder, one to predict the feature-wise mean of the latent distribution of the mask tokens, and another to predict the feature-wise variance of the latent distribution of the mask tokens. By doing this, we can enforce a KL-divergence loss (Kullback & Leibler, 1951) on the mask tokens to encourage them to come from a Gaussian distribution that is amenable to inference-time sampling, similar to the procedure for the VAE (Kingma, 2013).

Empirically, we also found it helpful to have a "fixed" mask token added to the pseudo-"mask" tokens from the variational encoder. This "fixed" mask token is still learnable, but it is fixed in the sense that it is not associated with the input image. We adopt the "fixed" mask token directly from He et al. (2022)'s implementation.

Finally, the variational encoder is only used at training time. At inference time or when transferred to other tasks, it is discarded, such that the model cost is the same as in the original MAE. This allows the model to be applied to masked or unmasked images: the deterministic encoder can process the visible portion in both cases, and for masked images, the unobserved tokens can be replaced with samples from a standard Gaussian distribution before passing through the decoder.

### 3.3 Model Formulation and Loss Function

We define notation for our model as follows: $D$ is the decoder; $E_d$ is the deterministic encoder; $\mathbf{x}_d$ is the "visible" portion of the image; $\mathbf{x}_m$ is the "masked" portion of the image; $\sigma(\mathbf{x}_m)$ is the standard deviation of the latent code from the variational encoder; and $\mu(\mathbf{x}_m)$ is the mean of the latent code from the variational encoder.

For model training, we wish to strike a balance between reconstruction ability and adherence of the masked tokens to the multivariate Gaussian distribution. As such, the loss function follows the beta-VAE (Higgins et al., 2017) loss function, except that the loss is only calculated on the "masked" tokens, *i.e.*, those that are fed into the variational encoder. The loss is as follows:

$$\mathcal{L}(\mathbf{x}_d, \mathbf{x}_m) = \text{MSE}\left(D(E_d(\mathbf{x}_d), \mu(\mathbf{x}_m) + \epsilon \odot \sigma(\mathbf{x}_m)), \mathbf{x}_m\right)$$
$$+ \frac{\beta}{2J} \sum_j^J \left(1 + \log(\sigma(\mathbf{x}_m)_j^2) - \mu(\mathbf{x}_m)_j^2 - \sigma(\mathbf{x}_m)_j^2\right). \tag{1}$$

where MSE is the mean squared error loss function; $\epsilon \sim \mathcal{N}(0, \mathbf{I})$; $j$ is the index of a feature in the latent code; $J$ is the total number of latent features across all tokens from $\mathbf{x}_m$; and $\beta$ is a hyperparameter that controls the weight of the KL-divergence. The first term (MSE) encourages the usefulness of representations for reconstruction, while the second term (weighted KL Divergence) encourages the adherence of the latent space to a multivariate Gaussian distribution amenable to sampling.

# 4 CONTEXT-COMPLETION: CONTEXTUAL INFERENCE BENCHMARK

The main difference between our method and the masked autoencoders of He et al. (2022) is the capability of our method, at inference time, to sample multiple possible inpaintings. Our first experiment presents way to evaluate whether the multiple inpaintings indicate better learning of the contextual relationships between objects in a scene. Given the same decoder architecture, the ability to give multiple plausible contextual inpaintings would be evidence of better representation quality.

## 4.1 EXPERIMENTAL SETUP

### 4.1.1 PRE-TRAINING PARTIAL VAE

We use the same encoder and decoder architectures as in He et al. (2022)'s MAE. For our experiments, we use the ViT-B encoder settings. With few modifications, we follow the hyperparameter choices and pre-training settings of He et al. (2022) when pre-training Partial VAE. We decrease the batch size to 1024 (from 4096). Secondly, since we have a new variational encoder, we set its $\beta$ (weight given to KL-divergence loss) to be 30, and set its learning rate to $\frac{1}{100}$ of that of the parameters in the rest of the model (to encourage learning stability).

### 4.1.2 MODELS COMPARED

The main comparison is between our Masked VAE and He et al. (2022)'s MAE, because at inference time, they have exactly the same encoder and decoder architectures (since we discard our variational encoder). This means that the difference in performance would be attributable to our method's enforcement of a multivariate Gaussian masked token space, as opposed to a fixed masked token.

We also benchmark against BEiT (Bao et al., 2021). We note that due to their use of DALL-E's discrete VAE, which was trained on 250 million images (Ramesh et al., 2021), to tokenize the images, it is not an apples-to-apples comparison. Furthermore, as originally presented, their work does not have contextual inpainting results, so we had to make some non-trivial modifications to their framework to enable it to do contextual inpainting: In essence, we combined the encodings of the visible tokens with the predicted mask token values and passed them through the previously neglected discrete VAE decoder. Details of our strategy can be found in Appendix A.

We do not benchmark against I-JEPA: it has some inpainting results in their paper (Assran et al., 2023), but these inpaintings are actually generated post-hoc via a separately trained diffusion model, which makes it difficult to evaluate the quality of the learned representation versus the contribution of this computationally-intensive decoder.

### 4.1.3 DATASET

For this new benchmark task, we use $3745$ images from the MS-COCO validation dataset (Lin et al., 2014), since it comes pre-annotated with the object classes and their locations. We filtered out images that only had bounding boxes that were too small (less than $10\%$ of the image area) or too large (over $85\%$ of the image area), since the inpainting task would become trivial or ill-posed.

### 4.1.4 METRIC CALCULATION

The aim of Context-Completion is to determine if the in-filled (previously masked) region matches any of the objects that could plausibly occur based on the other objects present in the context, *i.e.*, the visible portion of the image. Specifically, for each masked image, we calculate a binary vector

$$\mathbf{P}_i = \begin{cases} 1 & \text{if class } i \text{ could plausibly appear under the mask given } \mathcal{J} \\ 0 & \text{if class } i \text{ is } not \text{ plausible to appear under the mask given } \mathcal{J}, \end{cases} \quad (2)$$

where $\mathcal{J}$ is the set of the classes that appear in the visible context. Then, given $k$ inpaintings of this masked image from our model, we can calculate another binary vector

$$\mathbf{Q}_i = \begin{cases} 1 & \text{if class } i \text{ appears in any of the } k \text{ inpaintings} \\ 0 & \text{if class } i \text{ does } not \text{ appear in any of the } k \text{ inpaintings.} \end{cases} \quad (3)$$

| Method | Precision ($\uparrow$) | Recall ($\uparrow$) | Null ($\downarrow$) | Add Data |
|---|---|---|---|---|
| BEiT (Bao et al., 2021) | $0.0067 \pm 0.0814$ | $0.0002 \pm 0.0033$ | 0.9848 | yes |
| MAE (He et al., 2022) | $0.1208 \pm 0.3250$ | $0.0070 \pm 0.0294$ | 0.8611 | no |
| Masked VAE (Ours) | | | | |
| $k = 4$ | $0.3179 \pm 0.4511$ | $0.0248 \pm 0.0657$ | 0.5605 | no |
| $k = 8$ | $\mathbf{0.4110} \pm 0.4626$ | $\mathbf{0.0375} \pm 0.0833$ | **0.4109** | no |

Table 1: **Context-Based Object Inpainting Results:** We compare the performance of Masked VAE, MAE, and BEiT on our Context-Completion benchmark. Specifically, we inpaint contiguous objects on the MS-COCO (Lin et al., 2014) validation images. We sample $k = \{4, 8\}$ inpaintings from our model, whereas BEiT and MAE only give one. For **Precision** and **Recall**, the means and standard deviations of the metrics are presented across all validation images, for each model. **Null** is the proportion of images for which the model could not generate any detectable objects after $k$ samples. **Add Data** is whether the decoder used training data beyond ImageNet (Deng et al., 2009).

We then wish to see the class occurrence agreement between $\mathbf{P}$ and $\mathbf{Q}$. Note that $\mathbf{P}$ and $\mathbf{Q}$ are recalculated for each image, since each image contains different context objects $\mathcal{J}$. In our experiments, $k = \{4, 8\}$ for Masked VAE, $k = 1$ for MAE (since it has no sampling capabilities), and $k = 1$ for BEiT (as an image inpainting procedure was never defined in their original paper).

With this evaluation, we must consider a few things: (a) What metrics constitute class occurrence "agreement"? (b) How can we calculate which classes are plausible in $\mathbf{P}$? (c) How can we judge which classes are present during calculation of $\mathbf{Q}$?

In regards to (a), precision and recall are sensible metrics. In a given image, for each class $i$, we have a true positive when $\mathbf{P}_i = \mathbf{Q}_i = 1$, we have a false positive when $\mathbf{P}_i = 0, \mathbf{Q}_i = 1$, and we have a false negative when $\mathbf{P}_i = 1, \mathbf{Q}_i = 0$. We then calculate $precision = \frac{TP}{TP+FP}$ and $recall = \frac{TP}{TP+FN}$; where $TP, FP, FN$ are the number of classes that were respectively true positives, false positives, and false negatives in a particular image. (If both the numerator and denominator are $0$, we set the value to be $0$.) We then take the mean and standard deviation of the calculated precision and recall over all the images in the testing set. The idea is that precision captures how many of the inpaintings make sense, while recall reveals how well the inpaintings capture the true occurrence distribution. We also calculate the number of images for which the model could not generate any inpaintings after taking $k$ samples; that is, the number of images where $TP + FP = 0$. This shows the decoder's ability to generate anything resembling an object.

Regarding (b), we calculate plausible classes in $\mathbf{P}$ as follows. For each pair of classes $(i, j)$, we have a co-occurrence matrix $\mathbf{C}$ where $\mathbf{C}_{ij} = \mathbf{C}_{ji}$ is the number of images in the testing set where both $i$ and $j$ occurred. $\mathbf{C}_{ii}$ is the total number of testing images where class $i$ appeared, so $\forall i, j, \mathbf{C}_{ij} \leq \mathbf{C}_{ii}$. Unlike $\mathbf{P}$ and $\mathbf{Q}$, $\mathbf{C}$ is shared among all images. For a particular image, we calculate

$$P(i|\mathcal{J}) = 1 - \prod_{j \in \mathcal{J}} (1 - P(i|j)) = 1 - \prod_{j \in \mathcal{J}} \left(1 - \frac{\mathbf{C}_{ij}}{\mathbf{C}_{jj}}\right), \qquad (4)$$

where $P(i|\mathcal{J})$ is the occurrence probability of class $i$ in the masked portion of the image, conditioned on the visible classes. The calculation is the noisy-or operator (Pearl, 2014). Finally, to produce $\mathbf{P}$, we threshold $P(i|\mathcal{J})$, such that $\mathbf{P}_i = \mathbb{1}\left(P(i|\mathcal{J}) > \alpha\right)$. In our experiments we set $\alpha = 0.1$.

Finally, addressing point (c), there are multiple reasonable ways to do so. We calculate $\mathbf{Q}$ by using an off-the-shelf object detector (Faster R-CNN with ResNet50-FPN and V2 weights (Ren et al., 2016; Li et al., 2021) from PyTorch Model Zoo) on the inpainted part of the image only. We do not feed the whole image because we already know the classes present in the visible portion, and only need an automated way to determine which class is present in the inpainted portion. For any class $i$ that is detected with confidence above threshold $\beta$ in the inpainted region, $\mathbf{Q}_i = 1$; otherwise, $\mathbf{Q}_i = 0$. We set $\beta = 0.6$, and noticed that for many images, no class was detected with high confidence, in which case the $\mathbf{Q}_i$ remained 0.

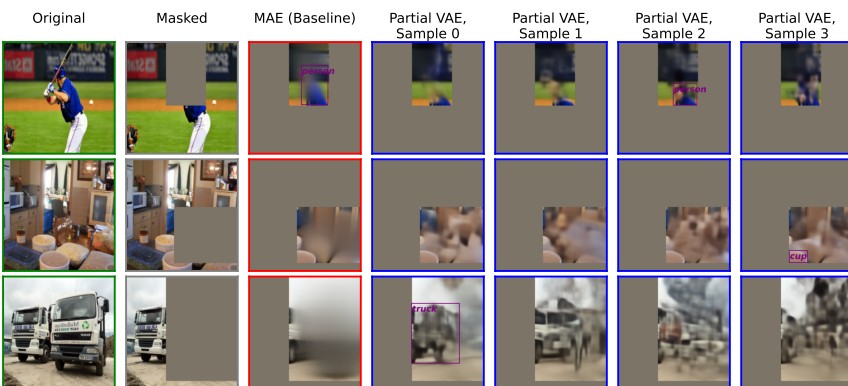

Figure 2: **Comparison of Object Detection in Context-Completion Benchmark:** Left-to-right: original uncorrupted image, masked image, inpainted portion by MAE (He et al., 2022), multiple inpainted portions from our Masked VAE. Data sourced from COCO validation set (Lin et al., 2014) via uniform sampling.

## 4.2 RESULTS

See Table 1 for results, and Figure 2 for some examples of detected objects. In precision and recall, our Masked VAE outperforms the standard MAE and BEiT at statistically significant levels, thereby validating that our sampling-based Masked VAE is superior at capturing contextual relationships.

There are also some peculiarities with the results. Firstly, the standard deviations (calculated over all the validation images) of the precision and recall are very high, compared to their mean values. This indicates that image-to-image, there is great variability in the ability of the methods to capture object co-occurrence relationships. Secondly, the recall results are abysmally low for all methods. This is somewhat expected, because given context objects, there is a wide variety of classes that could occur under the mask, and it would be unrealistic to expect a method to inpaint all of these objects in just $k = 4$ (for Masked VAE) or $k = 1$ (for MAE and BEiT) trials. Indeed, we see that the performance of Masked VAE increases when $k = 8$ as compared to $k = 4$ (remember that the baseline MAE and BEiT give $k = 1$ samples). Finally, in many cases, all models still cannot generate any detectable objects after $k$ samples. This occurs most frequently in the standard MAE and BEiT, and occurs less frequently as we increase $k$ for the Masked VAE.

## 5 END-TO-END FINETUNING EXPERIMENTS

### 5.1 EXPERIMENTAL SETUP

Following the setup of He et al. (2022), we evaluate the representation quality of Masked VAE via end-to-end fine-tuning on the ImageNet-1K training set and measuring accuracy on the validation set (Deng et al., 2009). As noted by He et al. (2022), this is preferable to linear probes for masked autoencoding methods, as the features learned may be strong albeit non-linear. This is especially relevant to the Masked VAE setup: due to the KLD penalty on the "masked" tokens, one could imagine that the training process may also indirectly encourage the "visible" tokens to follow a similar Gaussian distribution, since the decoder must process them all the same way. Accordingly, it may not be the case that the features are linearly separable, even if they are meaningful.

We use the *Deterministic Encoder* that was pre-trained on the "visible" tokens; we ignore the Variational Encoder that was pre-trained on the "mask" tokens. We use the hyperparameter settings provided by He et al. (2022)'s official implementation of fine-tuning their normalized pixel variant: https://github.com/facebookresearch/mae/blob/main/FINETUNE.md.

| Method | Arch. | Acc. |
|---|---|---|
| From Scratch (He et al., 2022) | ViT-B | 82.3 |
| DINO (Caron et al., 2021) | ViT-B | 82.8 |
| MoCo v3 (Chen et al., 2021) | ViT-B | 83.2 |
| BEiT (Bao et al., 2021) | ViT-B | 83.2 |
| MAE (He et al., 2022) | ViT-B | **83.3** |
| Masked VAE (Ours) | ViT-B | **83.3** |

Table 2: **End-to-End Finetune Results:** We show the comparison between Masked VAE and other leading self-supervised methods, as measured by top-1 validation set accuracy in end-to-end fine-tuning on ImageNet-1K (Deng et al., 2009). We see that Masked VAE is competitive with every method listed, when holding the backbone architecture fixed to ViT-B.

## 5.2 Results

In Table 2 we see that our results, when fixing the backbone to ViT-B (Dosovitskiy, 2020), are competitive with the existing methods, beating or matching (within rounding error) the performance of all methods (He et al., 2022). We also note that our method was trained for only 800 epochs, as compared to the 1600 reported by He et al. (2022); matching the number of epochs may improve the performance, although our invisible encoder incurs extra computational cost.

He et al. (2022) also explored a variant of the standard MAE with a reconstruction target of *per-token* normalized pixel values, as opposed to the regular raw pixel value loss (with normalization over the whole dataset). This variant, with a ViT-B backbone, achieved 83.6% accuracy, marginally above the 83.3% achieved by our Partial VAE and their version with regular pixel loss.

Furthermore, some methods, notably I-JEPA (Assran et al., 2023), only report results on larger ViT backbones which we did not have the computational resources to train. In this case, we can compare the performance to the standard MAE, which is perhaps the closest method to ours, using comparable backbones: I-JEPA with ViT-H/$16_{448}$ backbone achieves 87.1% accuracy, while the standard MAE with VIT-H/$14_{448}$ backbone and normalized pixel loss achieves 87.8% accuracy. The superiority of the standard MAE suggests that our method can be on par with I-JEPA on these large backbones.

## 6 Qualitative Results

In our Concept-Completion benchmar, we mask out contiguous objects as determined by the pre-annotated bounding boxes. This task is more challenging and requires more contextual understanding than random masking, since the model cannot simply interpolate between the nearest tokens. Figure 3 shows qualitative inpainting results on the COCO validation set (Lin et al., 2014), comparing our Masked VAE to the standard MAE (He et al., 2022).

We observe that the standard MAE has a tendency to almost linearly continue the color patterns of the bordering context into the masked region, even when this is not semantically meaningful. Furthermore, the inpaintings from the standard MAE are blurry and lack texture. In contrast, our Masked VAE is able to generate inpaintings with some semblance of texture, which makes the outputs look a bit more plausible. Furthermore, inherent to the sampling-based approach, our Masked VAE has diversity in its outputs (we show four sampled completions for each partial image), while the standard MAE can inherently only give one completion. That being said, both the standard MAE and the Masked VAE often give inpaintings that do not resemble objects, such as in the top two rows (bedroom, amusement park ride) of Figure 3. A larger decoder could improve sample quality, as could a GAN loss (Goodfellow et al., 2014).

## 7 Discussion, Limitations, and Conclusion

We introduced Masked VAE, a distributionally-aware self-supervised learning method that accounts for the inherent under-constrained nature of the masked vision pre-training paradigm. We accom-

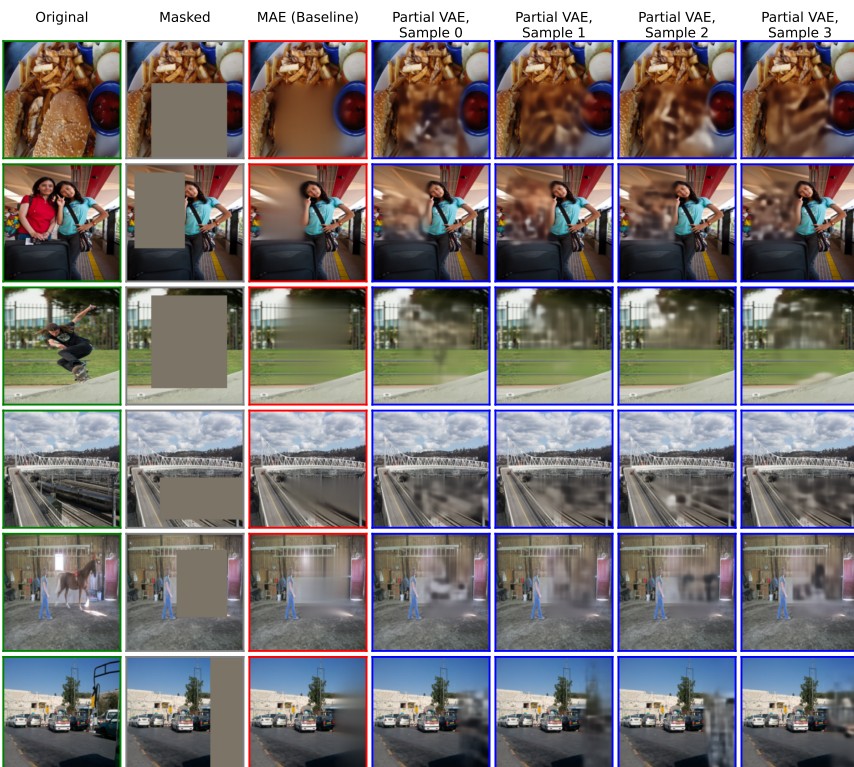

Figure 3: **Comparison of Inpainting Results:** Left-to-right: original uncorrupted image, masked image, inpainted image by MAE (He et al., 2022), multiple samples from our Masked VAE. Data sourced from COCO validation set (Lin et al., 2014) via uniform sampling.

plished this by associating possible image completions with samples from a multivariate Gaussian distribution in the latent space, while deterministically representing the visible image context.

We proposed Context-Completion, a new benchmark for context-aware representation learning: as a proxy for the ability of a representation to capture contextual relationships, it measures the ability to inpaint (with a fixed decoder architecture) a reasonable distribution of objects given a partial image. We quantitatively demonstrate that Masked VAE outperforms the standard MAE on Context-Completion. We also qualitatively examined the ability of Masked VAE to sample diverse meaningful completions of partial images. Finally, we showed that Masked VAE is a capable method for learning transferrable representations for classification, on par with the state-of-the-art given similar model resources.

Due to the extra encoder, our method inherently has more computational cost during pre-training time than the regular MAE has. However, we note that at inference time, whether for image inpainting with the decoder or transferring representations to downstream tasks, our method has no additional cost, since the variational encoder is discarded. Furthermore, the sample quality from our method has much room for improvement. We expect that using an adversarial loss (Goodfellow et al., 2014) or increasing the decoder size could significantly improve sample quality.

Beyond being a method for self-supervised learning, Masked VAE holds promise as a method for solving ill-posed inverse problems. A potential advantage over existing deep learning approaches such as diffusion models (Ho et al., 2020) is that the sampling is much quicker.

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

## A  APPENDIX

### A.1  BEIT MODIFICATIONS

As mentioned in Section 4.1.2, we had to make non-trivial modifications to BEiT (Bao et al., 2021) to enable it to do contextual inpainting in pixel space. Their pre-training task operates in a discrete token space that is created by DALL-E's pre-existing discrete VAE (dVAE) tokenizer (Ramesh et al., 2021) (in some sense making for two layers of autoencoding). Going beyond their original work, we take the extra step of using the dVAE's decoder to decode the completed image tokens after the original masked completion in token space. Specifically, we take the masked image, pass it through the dVAE encoder, and conduct the masked completion task in token space. We keep the encoded tokens that correspond to visible patches. Then, we simply concatenate the softmax probability distributions of the masked tokens with the visible tokens, and input them to the decoder. This is similar to the method used by Ramesh et al. (2021), although Bao et al. (2021) did not present any images generated via this method, to the best of our knowledge.

## A.2 BASELINE SOURCES

The baseline results in Table 2 are re-printed from He et al. (2022).

The standard MAE results in Table 1, Figure 3, and Figure 2 are generated with weights from: https://dl.fbaipublicfiles.com/mae/visualize/mae_visualize_vit_large.pth. The BEiT results in Table 1 are generated with ViT-B weights from: https://github.com/addf400/files/releases/download/v1.0/beit_base_patch16_224_pt1k_800ep.pth; and dVAE weights from https://conversationhub.blob.core.windows.net/beit-share-public/dall-e_vae/encoder.pkl and https://conversationhub.blob.core.windows.net/beit-share-public/dall-e_vae/decoder.pkl.

