# OpenReview forum: "Masked VAE: Distributionally-Informed Self-Supervised Vision Learning"
_ICLR.cc/2025/Conference — ICLR 2025 Conference Withdrawn Submission_

### Official Review · Reviewer_W8JF · 2024-11-01

**Soundness:** 2
**Presentation:** 3
**Contribution:** 2
**Rating:** 3
**Confidence:** 4

**Summary:**

This paper introduces the Masked Variational Autoencoder (Masked VAE), a self-supervised approach designed to learn contextually-aware representations of images. Specifically, the method employs a deterministic encoder and a variational encoder to process masked and visible image tokens, respectively. The combined features from both encoders are then used by a decoder to reconstruct the image. To assess the effectiveness of the proposed method, the authors establish a Context-Completion benchmark. Experimental results show that Masked VAE outperforms the baseline MAE on this benchmark

**Strengths:**

1. The paper is well-orgnized and easy to follow
2. Proposing contextually-aware representations introduces a novel perspective in visual representation learning. The methodology is intuitive and well-articulated.

**Weaknesses:**

1. The comparison between Masked MAE and the baseline MAE in the proposed Context-Completion benchmark appears somewhat unfair. MAE is primarily designed to encode general image representations with a lightweight decoder, which may not be suitable for image inpainting tasks. A more relevant comparison might be against specialized models designed for image completion, such as popular diffusion models.
2. The metrics used in the proposed benchmark seem outdated.
3. The demonstrated image completion results do not seem to match the performance of currently popular models in this domain.
4. The results presented in Section 5 suggest that the proposed method does not offer any improvements over the baseline MAE. This outcome is somewhat disappointing, especially for a model that builds upon the MAE framework.
5.  The motivation behind the paper is somewhat unclear, and the practical applications of the proposed method are not well defined. It would enhance the paper if the authors could clarify the intended use cases and potential impact of their approach within the field of self-supervised learning.

**Questions:**

N/A

---

### Official Review · Reviewer_WNpu · 2024-11-01

**Soundness:** 3
**Presentation:** 3
**Contribution:** 2
**Rating:** 3
**Confidence:** 4

**Summary:**

This paper proposes a MaskedVAE, a self-supervised learning method that builds upon the MAE by introducing distributional awareness.
This paper also introduces a new benchmark: Context completion, which is used for evaluating context-aware representation learning, by measuring the ability of inpainting objects given a partial image.
The authors show that MaskedVAE can be competitive with MAE on representation for classification, while outperforming MAE in context completion tasks, demonstrating the effectiveness of distributional awareness.

**Strengths:**

1. The motivation for introducing stochasticity to the masked prediction task is clear that the masked prediction task is under-constrained but existing works consider them as deterministic.

2. The paper is easy to follow.

3. MaskedVAE shows competitive performance to MAE on ImageNet fine-tuning, which indicates that introducing stochasticity in MAE does not harm the classification representation.

**Weaknesses:**

1. I think although masked prediction pre-text tasks do not involve stochasticity, their representation is quite good for understanding context, e.g., MAE achieves good performance on dense-prediction tasks, like object segmentation tasks. What is the benefit of context completion task rather than context understanding in representation space?

2. Similar to the 1, do we really need a good representation to in-paint the exact object even if it does not improve representation for any discriminative tasks? I think utilizing a much more powerful decoder like MAE (They used gan loss to fine-tune the model for more realistic generation) or adding diffusion models like I-JEPA may be enough to in-paint the masked part.

**Questions:**

1. Please answer the weakness part.

---

### Official Review · Reviewer_Emu9 · 2024-11-05

**Soundness:** 2
**Presentation:** 3
**Contribution:** 2
**Rating:** 3
**Confidence:** 4

**Summary:**

This paper proposes Mask VAE, which combines VAE and MAE to address the multiple hypotheses problem in masked image modeling tasks. The proposed method shows promising results on Context-Completion, a benchmark also introduced in this paper. This benchmark evaluates contextual modeling ability using object inpainting as a metric.

**Strengths:**

This paper reveals a problem in masked image modeling and proposes a probabilistic distribution from VAE to solve it. The benchmark demonstrates the superiority of this approach.

**Weaknesses:**

Although this paper presents a good problem and solution, the point I least understand is that solving this problem doesn't seem to bring any benefit to self-supervised learning: downstream tasks are only on par with MAE. Masked image modeling is just a **proxy task**, so if solving the flaws in this task doesn't benefit downstream tasks, is this flaw reasonable? If it's just for the image inpainting task, why wouldn't I use diffusion methods to solve it? Therefore, the position of this paper is not very clear.

**Questions:**

See weakness.

---

### Note · Authors · 2024-11-24

I have read and agree with the venue's withdrawal policy on behalf of myself and my co-authors.